Bark tissue transcriptome analyses of inverted Populus yunnanensis cuttings reveal the crucial role of plant hormones in response to inversion

Zhou An-Pei 1 2
Gan Pei-Hua 1 2
Zong Dan 1 2
Fei Xuan 1 2
Zhong Yuan-Yuan 1 2
Li Si-Qi 1 2
Yu Jin-De 1
He Cheng-Zhong hecz@swfu.edu.cn 1 2 3
1 Key Laboratory for Forest Genetic and Tree Improvement and Propagation in Universities of Yunnan Province, Southwest Forestry University , Kunming , China
2 Key Laboratory of Biodiversity Conservation in Southwest China, State Forestry Administration, Southwest Forestry University , Kunming , China
3 Key Laboratory for Forest Resources Conservation and Utilization in the Southwest Mountains of China, Ministry of Education, Southwest Forestry University , Kunming , China
Zhu Xiaocheng
Electronic publication date: 2019 Oct 1
Publication date: 2019
Volume: 7
Electronic Location ID: e7740
Received 2019 Apr 10; Accepted 2019 Aug 25
Copyright: ©2019 Zhou et al.
Copyright year: 2019
Copyright holder: Zhou et al.
License: This is an open access article distributed under the terms of the Creative Commons Attribution License, which permits unrestricted use, distribution, reproduction and adaptation in any medium and for any purpose provided that it is properly attributed. For attribution, the original author(s), title, publication source (PeerJ) and either DOI or URL of the article must be cited.
License URL: https://creativecommons.org/licenses/by/4.0/

Keywords: Bark tissues, Inversion, Transcriptome, Hub gene, Populus yunnanensis

Funding: National Natural Science Foundation of China 31860219 31360184 This work was supported by the National Natural Science Foundation of China (Nos. 31860219 and 31360184). The funders had no role in study design, data collection and analysis, decision to publish, or preparation of the manuscript.

==============================
Inverted cuttings of Populus yunnanensis exhibit an interesting growth response to inversion. This response is characterized by enlargement of the stem above the shoot site, while the upright stem shows obvious outward growth below the shoot site. In this study, we examined transcriptome changes in bark tissue at four positions on upright and inverted cuttings of P. yunnanensis: position B, the upper portion of the stem; position C, the lower portion of the stem; position D, the bottom of new growth; and position E, the top of new growth. The results revealed major transcriptomic changes in the stem, especially at position B, but little alteration was observed in the bark tissue of the new shoot. The differentially expressed genes (DEGs) were mainly assigned to four pathways: plant hormone signal transduction, plant-pathogen interaction, mitogen-activated protein kinase (MAPK) signaling pathway-plant, and adenosine triphosphate-binding cassette (ABC) transporters. Most of these DEGs were involved in at least two pathways. The levels of many hormones, such as auxin (IAA), cytokinin (CTK), gibberellins (GAs), ethylene (ET), and brassinosteroids (BRs), underwent large changes in the inverted cuttings. A coexpression network showed that the top 20 hub unigenes at position B in the upright and inverted cutting groups were associated mainly with the BR and ET signaling pathways, respectively. Furthermore, brassinosteroid insensitive 1-associated receptor kinase 1 (BAK1) in the BR pathway and both ethylene response (ETR) and constitutive triple response 1 (CTR1) in the ET pathway were important hubs that interfaced with multiple pathways.

Introduction

Plant polarity, an essential feature of differentiation along an axis of symmetry, designates the specific orientation of activity in space (Belanger & Quatrano, 2000; Qi et al., 2017). Studies in algae have suggested that an initial asymmetric division of the zygote plays a key role in the determination of different cell fates in early embryonic cells (Brownlee, Bouget & Corellou, 2001; Hable & Hart, 2010). The polar organization of molecules and cellular structures could significantly cause developmental changes in the morphogenesis, growth and function of cells and could allow specialized cell types to perform individual tasks (Belanger & Quatrano, 2000; Brownlee, Bouget & Corellou, 2001; Euchen et al., 2012). The flow of determination from the cellular to the tissue level predetermines the formation of various tissues with high polarity, such as root hairs and pollen tubes (Grebe, 2004).

Shoot-root polarity divides the plant body into two parts, the root at the base and the shoot at the top, and their apices point in opposite directions (Nick & Furuya, 1992). Baskin et al. (2010) proposed an apex-based terminology for plant polarity, including the new terms “shootward” and “rootward”, meaning specifically toward the shoot apex and toward the root apex, respectively. In higher plants, water and mineral elements are absorbed mainly by root hairs and are transported shootward into the aboveground parts, while organic nutrients are mainly photosynthesized in the leaves and transported rootward into the underground parts. This arrangement ensures the growth and development of plants. However, we found that certain inverted trees could still grow, in contrast to the typical case of woody plant polarity.

Yunnan white poplar (Populus yunnanensis), an important forest tree species from the Tacamahaca section of the Populus genus in the Salicaceae family, is a native dioecious species widely distributed in southwestern China (Chen et al., 2010; Jiang et al., 2012). Owing to their rapid growth, strong adaptability, easy asexual propagation, cold resistance, P. yunnanensis trees have been planted for the greening of cities and roads, and play an important role in forestry production, afforestation and environmental conservation (Jiang, Korpelainen & Li, 2013; Li et al., 2014; Ren et al., 2018). By chance, we found that inverted P. yunnanensis cuttings could form complete plants. The roots grew in the shootward direction, and shoots grew in the rootward direction; this development was linked to the physical orientation and not to the original polarity. The new shoots of inverted cuttings displayed less vigorous growth than did those of upright cuttings (Zhou et al., 2018). During the next year, obvious enlargement occurred above the shoot site of the inverted cuttings, while the upright cuttings exhibited outward growth below the shoot site (Fig. 1). This growth against plant polarity is of interest.

Figure 1 Vegetative form of P. yunnanensis cuttings.

(A) Upright cuttings. (B) Inverted cuttings.

Previous cases of shoot-root polarity were reported mainly during the 20th century (Bloch, 1943; Bünning, 1952; Sachs, 1969; Weisenseel, 1979; Nick & Furuya, 1992). Based on the morphogenesis of inverted plants, researchers have proposed either the hypothesis that polarity is induced de novo or the opposing view that polarity is stable. Vöhting’s experiment (Nick & Furuya, 1992) with inverted dandelion roots showed that adventitious shoots formed at the root pole when the shoot pole had been sealed with resin. However, a segment cut from an inverted plant without sealing the poles could grow shoots at the original shoot pole, which suggested an unaltered original polarity.

Inverted cuttings are considered excellent materials for understanding plant polarity, which is involved mainly in morphological reconstruction and shoot-root polarity. To date, relevant research has focused on how plant polarity is induced and fixed and how it orients cell division (Souter & Lindsey, 2000; Friml et al., 2006; Medvedev, 2012; Bringmann & Bergmann, 2017; Strzyz, 2017; Bornens, 2018). Although clear different from that of upright plant, the growth and development of inverted plants has largely been ignored. The different characteristics of the two direction types are the consequence of original shoot-root polarity changes, and some factors involved in polar responses play important roles in this process. In this study, inverted P. yunnanensis cuttings were characterized as having dwarf-type new shoots and enlarged stems above the shoot site. We performed transcriptome profiling of bark tissues in both upright and inverted cuttings and examined the main changes in the stems and new shoots in response to inversion. These findings could reveal the most important factors impacting the growth of inverted plants and could help to understand their action in response to plant polarity changes.

Materials and Methods

Plant materials

Three one-year-old P. yunnanensis clones from the Haikou Forest Farm of Kunming (China) were selected as three biological replicates, and their main stems were used to generate cuttings of similar length and diameter. We cultivated the cuttings at Southwest Forestry University (Kunming, China) during early spring (March) in two orientations: upright and inverted. With the exception of the middle bud on each cutting, all the buds were removed. During early August (the vegetative growth stage), the shoot branch sprouting from this bud and its cutting body were collected, and we sampled bark tissues at the following four positions (Fig. S1) to reveal any changes induced by inversion: positions B and C, which were above and below the shoot site, respectively; position D, which was at the bottom of the new growth; and position E, which was at the top. Four individuals with the same clonal origin, position and direction type were pooled to generate one mixed bark sample. A total of twenty-four bark samples from the four positions (three replicates each) were used to compare the transcriptome profiles of the two orientation treatments. The sample groups from the four positions (B, C, D, E) of the upright (U) and inverted (I) cuttings were named BU, CU, DU, EU, BI, CI, DI, and EI, and the three replicates within each group were numbered with 1 (e.g., BU1), 2 (e.g., BU2), and 3 (e.g., BU3).

RNA isolation and sequencing library preparation

Total RNA was isolated from each sample using an RNAprep Pure Plant Kit (Tiangen Biotech, Beijing, China). The RNA concentration and purity were subsequently assessed with a NanoDrop spectrophotometer (NanoDrop 1000; Thermo Scientific, Wilmington, Delaware, USA) and a Qubit fluorometer (Invitrogen, Carlsbad, CA, USA), and the RNA integrity was checked on an Agilent 2000 system (Agilent Technologies, Palo Alto, CA, USA). Each sample was used for cDNA library construction with a VAHTS Stranded mRNA-seq Library Prep Kit for Illumina (NR602, Vazyme, Nanjing, Jiangsu, China). In this step, the mRNA was purified from the total extracted RNA via oligo d(T) beads and was fragmented at 85 °C for 6 min. Random hexamers were used for first-strand cDNA synthesis, and second-strand cDNA synthesis and end repair were performed via Second Strand/End Repair Enzyme Mix. After purification via DNA Clean Beads, the double-stranded cDNA was processed with dA tailing and adapter ligation. The PCR amplification reaction was used to enrich the library. After quality control and quantification via an Agilent 2100 Bioanalyzer and a Qubit fluorometer, the qualified libraries were sequenced on an Illumina HiSeq X Ten device at 2 × 150 bp, and each library produced approximately 6 gigabases (Gb) of paired-end raw reads. The transcriptome data in this study were submitted to the short read archive (SRA) database under the accession number PRJNA506110.

De novo assembly and functional annotation

The raw Illumina reads were quality-trimmed and filtered to remove adapter sequences, reads containing poly-N sequences (>5%), and low-quality reads (Q < 20) by in-house Perl scripts. The clean reads were used for de novo assembly with Trinity (Grabherr et al., 2011). In brief, the short reads were first combined to generate linear contigs with a certain length of overlap based on K-mers values (K = 25). The contigs, which were comprised of alternative splicings and paralogous genes, were clustered to generate unigenes using the De Bruijn graph algorithm. The sequences were clustered to eliminate the redundant contigs using TGICL (Pertea et al., 2003). The generated unigenes were annotated using BLAST alignment tools (Altschul et al., 1990). A total of ten nucleotide and protein databases, namely, the non-redundant nucleotide sequences (NT), non-redundant protein sequences (NR), Swiss-Prot, clusters of orthologous groups of proteins (COG), gene ontology (GO), Kyoto encyclopedia of genes and genomes (KEGG), plant transcription factor database (PlantTFdb), plant resistance gene database (PRGdb), InterPro, and search tool for the retrieval of interacting genes/proteins database (STRINGdb), were used to determine the unigene functions. The parameters of these programs are listed in Table S1.

Detection of DEGs

We calculated the fragments per kilobase of transcript per million mapped reads (FPKM) to assess the expression levels of every sample using RSEM software (Li & Dewey, 2011). On the basis of read count, the differentially expressed genes (DEGs) between pairs of sample groups were detected using the DESeq package (Anders & Huber, 2010), and we chose those with a false discovery rate (FDR) ≤ 0.05 and a fold change (FC) ≥ 1. These DEGs (File S1) were further divided into two patterns: upregulation and downregulation. To assess the functions of the DEGs, a hypergeometric distribution test was applied to identify significantly enriched GO terms and KEGG pathways with TBtools (Chen et al., 2018), and their visualization was carried out with the ggplot2 package (Wickham, 2016).

Hub unigene search

When we compared the transcriptome differences between the control and treated samples, some hub unigenes were ignored by the DEG decision protocol. We thus used a weighted gene co-expression network analysis (WGCNA) (Langfelder & Horvath, 2008) to explore the coexpression of unigenes in highly enriched KEGG pathways, in which there were large numbers of DEGs. First, the soft power was determined via a soft-threshold approach, and the unigenes were clustered into coexpression modules. Then, the sample group information was imported, and the module-trait relationships were examined to search for key modules. Finally, the unigene interactions in these modules were visualized using Cytoscape software (Shannon et al., 2003; Cline et al., 2007), and a gene coexpression network based on hit number was constructed. Considering the expression information, we marked these unigenes and confirmed the top 20 hub unigenes per group. The Cytoscape working data are available in File S2.

Analysis of RT-qPCR

To validate the RNA sequencing (RNA-seq) results, we chose ten hub unigenes from the WGCNA and quantified them using real-time quantitative PCR (RT-qPCR). The RNA extracted for library preparation and transcriptome sequencing was used as a template for cDNA synthesis, and reverse transcription was performed using a FastQuant RT Kit (Tiangen Biotech, Beijing, China). EVA Green (Fast Super EvaGreen qPCR Master Mix; US Everbright, Inc., Suzhou, China) chemistry and a real-time PCR system (Rotor-Gene Q; Qiagen, Hilden, Germany) were used to validate the hub unigene expression. We used the endogenous control gene PD-EI (Pyruvate Dehydrogenase E1) to quantify the relative mRNA levels (Carraro et al., 2012; Yun et al., 2019). The primers of the hub unigenes were designed using Primer Premier 5 (Lalitha, 2000) and synthesized by Sangon Biotech Co., Ltd. (Shanghai, China). Information on all the primers used is listed in Table S2. Two-step amplification was carried out as follows: 95 °C for 2 min, followed by 40 cycles of 95 °C for 10 s and 60 °C for 30 s. The relative expression level was calculated by the 2−ΔΔCt method (Livak & Schmittgen, 2001). This experiment was repeated for three biological replicates, each involving three technical repeats.

Results

Transcriptome sequencing, de novo assembly and functional annotation

The RNA-seq libraries generated an average of 45,829,779 raw reads (40,920,000∼ 50,911,246). After quality control, an average of 44,153,404 clean reads were obtained from the 24 samples (39,078,392∼49,171,196). With Q20 >96.65% and Q30 >91.64%, more than 105 thousand unigenes per sample were assembled, and their N50 values were greater than 1436 (Table S3). These results indicated that the RNA-seq data were of high quality.

Among the 275,575 total unigenes, 208,387 (75.62%) were annotated to ten databases (Fig. 2), and 71% were aligned to Populus trichocarpa in the NR database (Fig. S2). This annotation proportion was relatively low, suggesting a number of low-reliability unigenes. We therefore filtered the unigenes from each sample according to a criterion of FPKM <1. As a result, there were a total of 204,840 unigenes, of which 174,264 (85.07%) corresponded to functional annotations (Fig. 2). After filtering, these unigenes were used as a basis for subsequent analyses.

Figure 2 Functional annotations of unigenes before and after filtering.

Figure 3 DEGs between upright and inverted cuttings.

BU indicates position B of upright cuttings. CU indicates position C of upright cuttings. DU indicates position D of upright cuttings. EU indicates position E of upright cuttings. BI indicates position B of inverted cuttings. CI indicates position C of inverted cuttings. DI indicates position D of inverted cuttings. EI indicates position E of inverted cuttings.

Detection of DEGs between upright and inverted samples

We compared the pairwise unigene expression changes in P. yunnanensis samples from the same or different positions in different orientations (Fig. 3) and found obvious pattern differences. A total of 9,590 and 6,785 DEGs were found in the pairs BU vs. BI and BU vs. CI, respectively, which were the largest numbers among all the pairs. At position C in the upright P. yunnanensis samples, 339 and 754 DEGs were identified in the pairs CU vs. CI and CU vs. BI, respectively. Few DEGs were found at the two positions in the new growth (D and E): 74 in DU vs. DI and 18 in EU vs. EI. These results suggested that the transcriptome changes induced by inversion were major in the older cutting regions but that there was little influence on the new growth.

The DEGs in four pairs (BU vs. BI, BU vs. CI, CU vs. CI, CU vs. BI) of cuttings were important for understanding the transcriptome changes (Table S4). The GO database was used to determine their enrichment, as shown in Fig. 4. Of the three main categories, biological process had the most GO terms in all four pairs (BU vs. BI: 24; BU vs. CI: 21; CU vs. CI: 17; CU vs. BI: 17), followed by cellular component (BU vs. BI: 16; BU vs. CI: 14; CU vs. CI: 11; CU vs. BI: 15). In the biological process category, metabolic process and cellular process were the most abundant terms. In the cellular component category, the most enriched terms were cell, cell part, membrane, organelle, and membrane part. In the molecular function category, many DEGs were related to binding and catalytic activity.

Figure 4 GO classification and term enrichment of DEGs.

(A) BU vs. BI. (B) BU vs. CI. (C) CU vs. CI. (D) CU vs. BI.

The DEGs were also subjected to KEGG annotation (Table S4). As shown in the KEGG enrichment results (Fig. 5), plant hormone signal transduction and plant-pathogen interaction contained the most DEGs in the pairs BU vs. BI and BU vs. CI, and they were significantly enriched in these two pathways. The pairs CU vs. CI and CU vs. BI had the most DEGs with significant enrichment in plant hormone signal transduction. All four pairs showed dramatic differences in plant hormone signal transduction. Among the DEGs in CU vs. CI, adenosine triphosphate-binding cassette (ABC) transporters, which are members of a transport system superfamily, were significantly enriched. Notably, the number of DEGs in mitogen-activated protein kinase (MAPK) signaling pathway-plants was always high, indicating a clear effect of inversion on this pathway.

Figure 5 KEGG pathway enrichment of DEGs.

(A) BU vs. BI. (B) BU vs. CI. (C) CU vs. CI. (D) CU vs. BI.

A number of DEGs were associated with two or more pathways (Fig. 6). In the plant hormone signal transduction pathway, most DEGs in BU vs. BI were related to auxin (also known as indole-3-acetic acid, IAA), cytokinin (CTK) and brassinosteroids (BRs) signaling, and the DEGs in CU vs. BI were assigned mainly to ethylene (ET) signaling. All four of these hormones, as well as gibberellins (GAs), were associated with a large number of DEGs in BU vs. CI. Most of these unigenes were upregulated in the pairs BU vs. BI, BU vs. CI and CU vs. BI, and the three biological replicates exhibited high homogeneity of expression.

Figure 6 DEG information in four pairs.

(A, B, C, D) BU vs. BI. (E, F, G, H) BU vs. CI. (I, J, K, L) CU vs. CI. (M, N, O, P) CU vs. BI. (A, E, I, M) Venn diagram of DEGs in four pathways: plant hormone signal transduction, MAPK signaling pathway-plant, plant-pathogen interaction and ABC transporters. (B, F, J, N) Number of unigenes assigned to orthology in hormone signaling. (C, G, K, O) Boxplot of expression levels of DEGs involved in hormone signaling. (D, H, L, P) Expression heatmap of DEGs involved in hormone signaling.

Gene coexpression network construction

We chose all the unigenes assigned to four KEGG pathways, including plant hormone signal transduction, plant-pathogen interaction, MAPK signaling pathway-plant, and ABC transporters, and grouped them into gene coexpression modules (Fig. 7). There was no clear outlier in the sample dendrogram. On the basis of an appropriate soft power of 10, the WGCNA divided these unigenes into 40 color modules. We examined these modules via the correlation between sample groups and gene modules and found that the hub modules were turquoise for BU, light green for BI, orangered4 and dark turquoise for CU, and pale turquoise for CI.

Figure 7 WGCNA of four KEGG pathways.

(A, B) Hierarchical clustering dendrogram of samples. (C, D) Scatter determining the soft threshold. (E) Hierarchical clustering of unigenes and module identification. (F) Network heatmap plot based on 500 randomly selected genes. The progressively more saturated yellow and red colors indicate high coexpression interconnectedness. (G) Relationships between gene modules and sample groups. (H) Hierarchical clustering of gene modules and sample groups. (I) Correlation heatmap of gene modules and sample groups.

The gene coexpression network including the DEG information revealed hub unigenes in each group (Fig. 8). The turquoise module was highly related to the BU group (0.904) and slightly related to the BI group (−0.060). Of the top 20 hub unigenes shown in Table 1, eight could encode brassinosteroid insensitive 1-associated receptor kinase 1 (BAK1), which is involved in BR signal transduction. Among the gene modules that were slightly related to the BU group, the light green module was most correlated with the BI group (0.756). Among the top 20 hub unigenes, a total of 16 were annotated with four compounds involved in the signal transduction of ET. These results indicated that some transcripts were inhibited and that some new functions were activated at position B when the cuttings were exposed to inverted conditions, and these changes approached significance. The correlation coefficients of the darkturquoise and orangered4 modules were high for the CU group but low for the CI group, and the correlation of the paleturquoise module was the opposite. A few genes in these three modules were differentially expressed.

Figure 8 Network of unigenes in modules highly connected to sample groups.

(A) Network of unigenes in the turquoise module, which is highly connected to the BU group. The unigenes aligned to the plant hormone signal transduction pathway are amplified to show detail. (B) Network of unigenes in the lightgreen module, which is highly connected to the BI group. (C) Network of unigenes in the darkturquoise and orangered4 modules, which are highly connected to the CU group. (D) Network of unigenes in the paleturquoise module, which is highly connected to the CI group. Each node indicates a gene, and its size indicates the number of hits to this gene. The log2FC values of the unigenes are characterized by five colors: green, showing log2FC <  − 2; blue, showing −2 ≤ log2FC <  − 1; pink, showing −1 ≤ log2FC ≤ 1; orange, showing 1 < log2FC ≤2; and red, showing 2 < log2FC. The unigene names are provided when Q values < 0.05, and two sizes are used based on different significance levels: the larger indicates a significant expression change at the 0.01 level, and the smaller size indicates a significant expression change at the 0.05 level.

Table 1 Information on the top 20 hub unigenes. “NA” indicates that the unigenes were not involved in the plant hormone signal transduction pathway.

Gene module	Plant hormone signal transduction pathway	KEGG orthology	KO ID	Number of unigenes	
Turquoise	Brassinosteroid	BAK1	K13416	8	
	Abscisic acid	PP2C	K14497	4	
	Jasmonic acid	MYC2	K13422	4	
	Ethylene	CTR1	K14510	2	
		SIMKK	K13413	1	
	Salicylic acid	PR-1	K13449	1	
Lightgreen	Ethylene	ETR	K14509	8	
		EIN3	K14514	4	
		EBF1/2	K14515	3	
		CTR1	K14510	1	
	Abscisic acid	PP2C	K14497	1	
	NA	NA	NA	3	

RT-qPCR verification

The gene expression levels of ten hub unigenes (Table S5) obtained from the WGCNA results were validated via RT-qPCR (Fig. S3). Of these genes, six were downregulated, and four were upregulated. They all matched well with the RT-qPCR results, which corroborated the reliability of the RNA-seq results.

Discussion

BRs and ET are important hubs for changes in hormone signaling

Plants coordinate their growth and development to adjust to the external environment (Santner & Estelle, 2009). This adjustment is a complex biological process in which hormones play a role. Plant hormones, including IAA, CTK, GAs, abscisic acid (ABA), ET, BRs, jasmonic acid (JA), and salicylic acid (SA), function in various aspects of growth, either singlehandedly or interactively (Santner, Calderon-Villalobos & Estelle, 2009; Band et al., 2012; Pacifici, Polverari & Sabatini, 2015). This complexity arises from hormone biosynthesis, transport, and signaling pathways and from the diversity of interactions between hormones. In the bark transcriptome analysis performed in this study, a number of DEGs were annotated to the plant hormone signal transduction pathway, including all the hormones in this map, which indicated a large influence of inversion on hormone signaling pathways. The coexpression network also showed that the top 20 hub unigenes were mainly part of two hormone signaling pathways: those involving BRs and ET.

BRs are major growth-promoting steroid hormones that modulate cell elongation and division (Mandava, 1988; Clouse & Sasse, 1998; Belkhadir & Jaillais, 2015; Vragovic et al., 2015). In addition to playing important roles in root growth and development, BRs actively function in stem elongation and vascular differentiation (Singh & Savaldi-Goldstein, 2015; Ahanger et al., 2018). BR-insensitive mutants of Arabidopsis thaliana are characterized by multiple deficiencies in developmental pathways, and these mutants exhibit severe dwarfing, dark green color, thickened leaves, and reduced apical dominance (Clouse, Langford & McMorris, 1996; Bishop, 2003). BRs have also emerged as crucial regulators of the growth-immunity trade-off (Krishna, 2003; De Bruyne, Höfte & De Vleesschauwer, 2014; Lozano-Duran & Zipfel, 2015). Serving as a key mediator of environmental stress factors, ET is a gaseous hormone involved in many developmental processes and responses to biotic and abiotic stresses in plants (Corbineau et al., 2014; Müller & Munné-Bosch, 2015; Thao et al., 2015; Broekgaarden et al., 2015; Berrabah et al., 2018). In this study, these two hormones were associated with hub unigenes detected in the bark tissues of the inverted cuttings of P. yunnanensis. The top 20 hub unigenes in the BU and BI groups were involved mainly in the BR and ET signaling pathways, respectively. Furthermore, among the constituents of the BR signaling pathway, only BAK1 was identified, and this protein could be encoded by 8 of the top 20 hub unigenes. In the BI group, 16 of the top 20 hub unigenes were related to ET signaling. The ethylene response (ETR) protein was associated with the most unigenes (8). These results indicated that the main changes induced by inversion occurred in the BR and ET signaling pathways.

Moreover, the signaling pathways of other hormones in bark tissues also strongly impacted the growth and development of the inverted cuttings of P. yunnanensis. Multiple hormones are at play, and the cooperation and crosstalk between their signaling pathways are complex (Depuydt & Hardtke, 2011), as reflected by the coexpression network in this study. Because of the typical triple response, ET is related to radial swelling of the stem, which could result in obvious enlargement above the shoot site of inverted cuttings. Various studies (Muday, Rahman & Binder, 2012) have suggested that, by altering the signaling, synthesis and transport of IAA, ET affects many aspects of IAA-dependent seedling growth. Among all the hormones, IAA, which was enriched for most DEGs, is a peculiar plant hormone because of its own polar transport route and functions in the proliferation and elongation of cells (Santner & Estelle, 2009; Tian et al., 2018), which could contribute to the enlargement of the stem of inverted cuttings. Several researchers have hypothesized that the polarity in inverted flowering plants cannot be reversed and that cuttings are resistant to changes in the polar direction of IAA transport (Sachs, 1969; Nick & Furuya, 1992; Friml et al., 2006). Despite these hypotheses, new downward flow seems to occur in addition to the original upward polarity (Went, 1941). The clear increase in outward growth below the shoot site of inverted cuttings (Fig. 1) supported the hypotheses concerning new polar IAA. BRs play an important role in vascular differentiation (Fukuda, 2004; Caño Delgado et al., 2004), which may induce the establishment of new cambium tissue in inverted plant cuttings to transport new IAA in the opposite direction in addition to old cambium that retains its IAA polarity.

Hub unigenes also function in MAPK signaling and plant immunity

Cutting inversion notably affected the MAPK signaling pathway and plant-pathogen interactions. A number of constituents in these two pathways were associated with the DEGs. For clear identification, we focused on the top 20 hub unigenes obtained from the WGCNA and coexpression network. In addition to their important roles in plant hormone signal transduction, many of these unigenes also actively function in MAPK signaling (all 20 unigenes) and plant immunity (10 unigenes), which indicates that close links among these three pathways and an important hub consisting of a few genes annotated to these pathways exist.

Some genes involved in hormone signal transduction such as BAK1 can function in the MAPK pathway and in plant-pathogen interactions (Li et al., 2002; Chinchilla et al., 2007). BAK1, which is also known as BAK1/SERK3, is the third member of the small somatic embryogenesis receptor kinase (SERK) family of Arabidopsis; it is a receptor-like kinase implicated in plant immunity and MAPK signaling (Nam & Li, 2002; Heese et al., 2007). Recognition of microbe/pathogen -associated molecular patterns (MAMPs/PAMPs) are central to innate plant immunity (Yamada et al., 2016; Yasuda, Okada & Saijo, 2017). In the MAPK pathway, the proteins BAK1 and Flagellin Sensing 2 (FLS2) form a receptor-like kinase complex located in the plasma membrane (Chinchilla et al., 2007). This complex recognizes a 22-amino acid peptide from flagellin (Flg22), a major structural protein of the eubacterial flagellum that acts as a PAMP in plants (Colcombet & Hirt, 2008; Meng & Zhang, 2013). In the plant-pathogen interaction pathway, BAK1 is recruited to FLS2 and elongation factor (EF)-TU receptor (EFR), which can recognize the bacterial MAMP flg22 epitope and EF-TU (elf18 epitope), respectively (Heese et al., 2007; Henry, Yadeta & Coaker, 2013). When BAK1 is silenced, Nicotiana benthamiana exhibits attenuated resistance to bacterial and oomycete pathogens (Chaparro-Garcia et al., 2011). Therefore, BAK1 is an important hub that functions in various signaling pathways, and it was also identified as a hub in the altered bark transcription network induced by inversion in P. yunnanensis.

Plant ET hormone signaling also involves MAPK cascades (Chang, 2003; Ouaked et al., 2003; Schweighofer & Meskiene, 2008; Li et al., 2018), which are highly conserved signaling pathways across eukaryotes. Membrane-localized ETRs are similar to bacterial two-component histidine kinases and are encoded by ETR1, ETR2, ethylene resistant 1 (ERS1), ERS2, and EIN4 in Arabidopsis (Schweighofer & Meskiene, 2008). Previous studies have revealed that ETRs are negative regulators and actively repress downstream components such as constitutive triple response 1 (CTR1), a MAPK kinase kinase (MAPKKK) that acts as a negative regulator in ET signaling (Hua & Meyorowitz, 1998; Qu et al., 2007; Bakshi et al., 2015). In this study, ETR and CTR1 were encoded by 8 and 1 of the top 20 hub unigenes in the BI group, respectively, which indicated a strong effect on the receipt and transmission of ET after inversion in P. yunnanensis cuttings.

However, the genes related to plant immunity might not be directly related to polarity. A polar signal induced by inversion, which is similar to induction caused by external stress, might activate the immune response mechanism of plants. The genes related to hormone signaling first responded intensely to inversion, which also resulted in the activation of plant immunity as an incidental to hormone alteration because of sharing among common hub unigenes. This great attention to the “stress” could limit the growth and development of inverted plants. Despite a lack of actual chemical detection, it is believed that hormones play a crucial role in responding to inversion, but additional experiments are needed to verify the involvement of hormones in polarity change.

Conclusions

Cuttings of P. yunnanensis present an interesting growth response to inversion, characterized by enlarged stems and dwarf-type new shoots. Our study focused on transcriptome changes in bark tissue induced by cutting inversion. The results revealed the major transcriptome changes induced by inversion in the older cutting regions, but there was little influence on the new growth. Moreover, plant hormones played a crucial role in the inverted cuttings. The levels of many hormones, including IAA, CTK, GAs, ET, and BRs, underwent large changes in the inverted cuttings. Furthermore, BAK1 in the BR pathway and ETR and CTR1 in the ET pathway were important hubs that interacted with multiple pathways, such as MAPK signaling and plant immunity.

Supplemental Information

Figure S1 Sampling positions on the cuttings of Populus yunnanensis

Click here for additional data file.

Figure S2 BLAST results of unigenes in the NR database

(A) E-value distribution statistics. (B) Similarity distribution statistics. (C) Species distribution statistics.

Click here for additional data file.

Figure S3 RT-qPCR verification of ten selected hub unigenes in comparison with the transcriptome results

The normalized RT-qPCR data are given as the means ± standard errors (SEs) of three biological replicates.

Click here for additional data file.

Table S1 Parameters of the programs used in the de novo assembly and functional annotation

Click here for additional data file.

Table S2 Information on ten selected hub unigenes and endogenous control gene used for RT-qPCR analysis

Click here for additional data file.

Table S3 Statistics of sequencing data and assembly results before filtering

Click here for additional data file.

Table S4 Function annotation of unigenes based on GO and KEGG databases

Click here for additional data file.

Table S5 RT-qPCR verification of the transcriptome results

Click here for additional data file.

File S2 Cytoscape working files

Click here for additional data file.

File S1 Cytoscape working files

Click here for additional data file.

Supplemental Information 3 The raw data of RT-qPCR analysis

The RT-qPCR raw data file shows the Ct values of samples and control gene PDEI. The relative expression level was calculated by the 2−ΔΔCt method.

Click here for additional data file.

We thank Prof. Aizhong Liu, Kunming Institute of Botany, for his help with the experimental design.

Additional Information and Declarations

Competing Interests

Author Contributions

Data Availability

The authors declare there are no competing interests.

An-Pei Zhou conceived and designed the experiments, performed the experiments, analyzed the data, prepared figures and/or tables, authored or reviewed drafts of the paper, approved the final draft.

Pei-Hua Gan analyzed the data, prepared figures and/or tables.

Dan Zong, Xuan Fei, Yuan-Yuan Zhong and Si-Qi Li performed the experiments.

Jin-De Yu approved the final draft.

Cheng-Zhong He conceived and designed the experiments, approved the final draft.

The following information was supplied regarding data availability:

The transcriptome sequences are available at Figshare: Zhou, Anpei (2019): transcriptome sequences of Populus yunnanensis.fa. figshare. Dataset. https://doi.org/10.6084/m9.figshare.7973858.v2.

The sequences are available at GenBank: https://www.ncbi.nlm.nih.gov/search/all/?term=PRJNA506110.

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
