# Peer review of "Bark tissue transcriptome analyses of inverted Populus yunnanensis cuttings reveal the crucial role of plant hormones in response to inversion"

_PeerJ, doi:10.7717/peerj.7740_

## Round 0.1 · original submission · Minor Revisions

Both reviewers gave very positive comments on your manuscript. In addition to those points, I also have a few comments for you to consider:
1. There is no chemistry data to show the level of hormones changes. The RNA seq data did show the changes in the level of expression of genes related to the passway. However, it doesn't guarantee that the actual change happened (highly possible though). If you don't have the capacity to do the chemical analysis, at least point this out in the discussion and possibly as future work.
2. The research itself is interesting, however, how your research can be applied to? why it is useful?
3. What's after that? Any suggestions for future research points?
4. You need to add the affiliation for Prof. Aizhong Liu in the acknowledgment.

[]

·

Basic reporting

- English must be improved.
- References provided but some important refrences must be added for the used programs.
- Too much figures
- Research hypothesis and specific objective must clearly detailed

Experimental design

The experimental set up is clear.
Research questio must be specified in details at the end of the introdution
More details must be added to the used methods

Validity of the findings

Data re robust.
Conlusions well stated

Additional comments

The idea and the set up of the manuscript The manuscript ''Bark tissue transcriptome analyses of inverted Populus yunnanensis cuttings reveal the crucial role of plant hormones in responding to inversion'' are very interesting. The different part of the manuscript were welle structured and illustrated. However, the English should be checked in some parts. Also, some important details are lacking and should be clarified for the readers.

Comments

Introduction
1 - The research hypothesis and the specific objectives of the work should be clearly detailed in the introduction
2- Lines 44-45. The authors should explain what they mean by 'developmental significance'.
3 - The authors should provide the name of the botanical family of Populus yunnanensis. Some details about the ecological and economical value may be also interesting to justify the choice of the species in this study.
Materials and Methods
4- Line 87: the word 'cultured' must changed by 'cultivated'
5- Details about the geographical origin of the samples must be provided
6- For RNA-seq, we assume that mRNA was purified from the total extracted RNA and then used for cDNA preparation. Please provide details about the protocol of mRNA purification (I guess with oligodt beads).
7- Details about the steps followed for cDNA preparation must be provided (mRNA fragmentation, reverse transcription, first and second strand preparation, adaptors ligation ...)
8- Details about the method used for quantify the cDNA libraries must provided. This is an important step before proceeding to the sequencing.
9- Please specify that the sequencing performed is a pair - end (2 × 150 bp)
10- The authors should add details about the quality of the generated transcriptome. Several approaches could be used
- CD-HIT program for contigs clustering and elimination of redundant contigs
- BUSCO program to align the generated transcriptome with provided databases (in your case; Plants or eukaryotes databases)
- maping back each sample to the generated transcriptome and check the % of similarity
- For all databases used for annotations, programs, parameters used and references must provided.
Results
- More details must provided in the part ' Real-time quantitative PCR verification'. The authors should give value to the results of RT PCR analysis and compare them with the results of RNA-seq approach.
Discussion
The discussion part was well prepared.
- Too much figures in the manuscript! Some of them may be combined in one figure to reduce the total number of the figures.

·

Basic reporting

The manuscript "Bark tissue transcriptome analyses of inverted Populus yunnanensis cuttings reveal the crucial role of plant hormones in responding to inversion" (#36443v1) by Zhou et al. is a very interesting work concerning the transcriptome analysis of the response to inversion in bark tissues of this species.Professional Engilsh is used throughout the manuscript. The work is well planned, and the technical development of the analysis is excellent, making use of many bioinformatics tools to properly examine the results.Figures are adequately designed and very informative. Authors have generated a vast amount of information that is very useful.I find the work suitable for publication in its current state.

I will only make some minor comments and suggestions:

- The discussion, though correct and well referenced, could have been more sound. In the introduction the authors write about plant polarity, but this issue is not further discussed concerning the results. Indeed, the involvement of the ethylene in the lightgreen module (related to BI) provides interesting information about the implication of this hormone in polarity. In a similar trend, the putative relation of plant-pathogen responsive genes with polarity is neither discussed. Nonetheless, I understand that inversion is a neglected issue in plant physiology and there are not many references about it.

- There is no information about the primers of the PEDI gene.

- Line 311: Nicotiana benthamiana. Names of species should always be in italics. Besides, some references are in italics while others not.

- Line 484: Estelle M.

Experimental design

No comment.

Validity of the findings

No comment.

---

## Round 0.2 · Minor Revisions

Please find below comments from Gerard Lazo, the Section Editor:

“The fact that a classification of expression by a collection of annotation methods was encouraging and appeared useful for the understanding of the gene categories involved; however, there is no way for the reader to link this information to the sequences provided (it is only seen as a list of annotations in a figure without the corresponding terms (e.g. GO12456). There were 275575 unigenes listed in the supplemental data; perhaps a table listing the associated GO and KEGG terms with each of these would be appropriate, or to add them to the annotation description for the FASTA sequences provided. Other opportunities to assist the reader would be to provide the Cytoscape working files so that visualization can be re-created. A key role in presenting data is to let the reader become aware of it for validation rather than relying solely on the word of the authors. In many of these cases, a change in parameters may help to yield different viewpoints. Journal manuscripts are often scanned by text-mining software that locates and extracts core data elements, like gene function. Adding standard ontology terms, such as the Gene Ontology (GO, geneontology.org) or others from the OBO foundry (obofoundry.org) can enhance the recognition of your contribution and description. This will also make human curation of literature easier and more accurate. The manuscript would be ready once the annotation-links can be created; until then, this should be placed into a ‘Minor Revision’ classification.”

Other minor changes:

Line 173 RT-qPCR
Line 190 "de novo" need to be italic
Line 206 9,590 and 6,785
Line 387 delete the acknowledgement for editor and reviewers

---

## Round 0.3 · accepted · Accept

Please revise the following minor point before publication:

1. The new supplementary files (Cytoscape working files and DEGs detection) need to be cited in your manuscript. Maybe in the Materials and Methods.